# Exploring providers' perceived barriers to utilization of antenatal and delivery services in urban and rural communities of Ebonyi state, Nigeria: A qualitative study

Pearl Chizobam Eke[1], Edmund Ndudi Ossai[2,3]*, Irene Ifeyinwa Eze[2,3], Lawrence Ulu Ogbonnaya[2,3]

1 Department of Nursing Services, Alex Ekwueme Federal University Teaching Hospital Abakaliki, Abakaliki, Nigeria, 2 Department of Community Medicine, College of Health Sciences, Ebonyi State University Abakaliki, Abakaliki, Nigeria, 3 Department of Community Medicine, Alex Ekwueme Federal University Teaching Hospital Abakaliki, Abakaliki, Nigeria

* ossai_2@yahoo.co.uk

**Data Availability Statement:** All relevant data are within the paper and its Supporting information files.

## Abstract

### Objective

To determine providers' perceived barriers to utilization of antenatal and delivery services in urban and rural communities of Ebonyi state, Nigeria.

### Methods

A descriptive exploratory study design was used. Qualitative data was collected through the use of a pre-tested interview guide. Twelve providers participated in the study in urban and rural communities of Ebonyi State, Nigeria. They included nine officers in charge of primary health centers, two Chief Nursing Officers of a tertiary health institution and mission hospital and one Medical Officer-in-charge of a General hospital. QDA Miner Lite v2.0.6 was used in the analysis of the data.

### Results

Most providers in urban and rural communities attributed good utilization of maternal health services to delivery of quality care. Most providers in urban linked poor utilization to poor health seeking behavior of women. In rural, poor utilization was credited to poor attitude of health workers. Few of participants (urban and rural) pointed out the neglect of primary health centers resulting in poor utilization. Most participants (urban and rural) considered ignorance as the main barrier to using health facilities for antenatal and delivery services. Another constraint identified was cost of services. Most participants attested that good provider attitude and public enlightenment will improve utilization of health facilities for antenatal and delivery care. All participants agreed on the need to involve men in matters related to maternal healthcare.

**Funding:** The authors received no specific funding for this work.

**Competing interests:** The authors have declared that no competing interests exist.

## Conclusions

Participants were aware of values of good provider attitude and this is commendable. This combined with the finding of poor attitude of health workers necessitates that health workers should be trained on quality of care. There is need for public enlightenment on need to utilize health facilities for antenatal and delivery services. Community ownership of primary health centers especially in rural communities will enhance utilization of such facilities for maternal healthcare services and should be encouraged. Involvement of men in matters related to maternal healthcare may have a positive influence in improving maternal health in Nigeria.

## Introduction

For over a century, the death of a woman while pregnant or during puerperium has been regarded as a public health problem [1]. This problem is more pronounced in developing countries. For instance, in 2017 the sub-Saharan African region accounted for about 66% of all maternal deaths in the world [2]. Also, the life time risk of maternal death in sub-Saharan Africa was 1 in 37 and this is very high when compared with that obtained for Australia and New Zealand which was 1 in 7800 [2]. On the country level, Nigeria bears the highest burden of maternal deaths globally [2]. There is evidence that the country also has one of the largest burdens of obstetric fistula in the world and this is a complication of pregnancy [3]. The current maternal mortality ratio for Nigeria is estimated at 512 maternal deaths per 100,000 live births and the burden is higher in rural areas when compared to urban [4].

The use of antenatal care and skilled attendant at delivery have been identified as capable of improving child survival and maternal health [2, 5, 6]. Thus, the observed differences in maternal mortality ratio between developed and developing countries could be attributed to the different rates of utilization of modern obstetric services, such as antenatal care (ANC) and skilled delivery care. In Nigeria, the utilization of maternal health services is higher in urban when compared to rural areas [4]. Suffice it to say that the use of these services is complementary as women who attended ANC four or more times have increased odds of delivering in a health facility [7, 8]. To a large extent, ensuring health facility delivery by pregnant women in Nigeria is capable of improving the high maternal death burden in the country and this should be encouraged.

Unfortunately, in most developing countries, the use of antenatal and skilled birth attendant is low and this includes Nigeria. For example, from the Nigerian Demographic and Health Survey, 57% of pregnant women in Nigeria made four or more antenatal care visits while 43% of births were attended to by skilled birth attendants [4]. It is important to note that all international efforts aimed at reducing maternal morbidity and mortality including Safe Motherhood Initiative, Millennium Development Goals and presently the Sustainable Development Goals (SDG) amongst others have all emphasized the need for good utilization of antenatal and delivery services. This brings to the fore the need to increase the utilization of these services by pregnant women in Nigeria. The SDG 3 aims to reduce global maternal mortality ratio to 70 maternal deaths per 100,000 live births by 2030. This has been interpreted as having a vision of putting an end to all preventable maternal deaths globally [2]. Consequently, there is need for Nigeria to increase its utilization of antenatal and delivery services not only to move in tandem with global trend but also to decrease the high maternal death burden which is the highest in the world at present.

A systematic review on access barriers to maternal health in low-income countries in Africa from 2015 revealed that lack of transportation to health facilities, economic factors and cultural

beliefs were the main barriers to maternal health from perspective of community members and health providers [9]. The results of a study involving health providers and community key informants identified irregular and poor-quality services, inadequate human resource for health and poor governance as the health system factors militating against the use of child birth services in a rural setting in Nepal [10]. Another study in Nepal identified inadequate knowledge of services offered by skilled birth attendants, distance to health facilities and unavailability of transport services as major barriers to use of skilled birth care [11]. A study in Malawi identified delayed health care financing, inadequate human resource for health and poor record keeping as the barriers associated with maternal health care service delivery [12]. From the results of a study in Ethiopia, health providers were of the opinion that women deliver at home due to inadequate resources in health facilities and the perception by the women that delivery in health facilities is unappealing amongst other factors [13].

The providers of maternal health services have been identified as playing prominent roles in the use of maternal health services and also the satisfaction of the women with services received [14, 15]. This has necessitated the call that health workers should be trained on quality health care [14, 15]. There has been a postulation that in decreasing the high maternal mortality in Nigeria that attention should be paid to inhabitants of rural areas [14], where the burden of maternal deaths is higher [4]. It has also been emphasized that in the years beyond the Millennium Development Goals, that the key to making informed decisions towards reducing maternal mortality requires understanding of both the drivers and inhibitors of progress towards reducing the death of women during pregnancy and puerperium [16]. Furthermore, a point has been made on the need to understand factors that affect the use of maternal health services at the community level so as to be able to develop interventions that will adequately suit every community [17]. In this context, even though the utilization of skilled providers for antenatal and delivery services in southeast Nigeria is the highest among the six geo-political zones in the country, the records for Ebonyi state are the least in the zone [4]. This calls for concern and the need for improvement. This study was therefore designed to determine providers' perceived barriers to utilization of antenatal and delivery services in urban and rural communities of Ebonyi state, Nigeria.

## Methods

### Study setting

The study was conducted in Ebonyi State, one of the five states in southeast geo-political zone of Nigeria. The state was created out from the old Abia and Enugu States by October 1996 and its administrative capital is Abakaliki. The inhabitants are mainly of Igbo ethnic nationality. Ebonyi State has 13 local government areas, (LGA) three are classified as urban while the remaining ten are regarded as rural. Ebonyi state like other states in Nigeria operates a healthcare system which is based on primary health care at the base supported by secondary and tertiary healthcare levels. There are 545 health facilities in the state including 530 primary healthcare facilities, 13 secondary healthcare facilities and two tertiary health institutions. When compared with other states in southeast Nigeria, Ebonyi state has the least utilization of antenatal and delivery services with a skilled attendant. The state also has the worst maternal health index in the zone [4].

### Study design, participants and sampling

This was a qualitative descriptive study. Information was obtained using a pre-tested interview guide for providers of antenatal and delivery services in urban and rural communities of Ebonyi state, Nigeria. Twelve providers of antenatal and delivery services participated in the key

informant interview (KII). Half of the providers serve in urban area. Nine were Officers in Charge of primary health centers while two were Chief Nursing Officers of a tertiary health institution and a mission hospital. One participant was a Medical Officer in charge of a General hospital. They were selected based on their official positions and the roles they play in policy formulation and in the delivery of maternal health services in the health facilities.

A two-stage sampling method was used for the study. In the first stage, a simple random sampling technique of balloting was used to select two local government areas each in urban and rural areas. There are three local government areas in urban and ten in rural areas of the state. In the second stage, a list of all health facilities in the selected local government areas was made and ranked based on the number of women that utilized the health facilities for antenatal and delivery services in the last six months based on hospital records. Twelve health facilities, six each in urban and rural communities were selected based on this criterion. The key providers of antenatal and delivery services in each of the facilities were purposively selected for interview.

## Study instrument and data collection methods

A pre-tested key informant interview guide was used to obtain information from the providers. Five KII guides were pretested among providers of ANC and delivery services in another local government area not selected for the study. The aim of the pretesting was to detect deficiencies or ambiguities of the study instrument and necessary corrections were made when they were detected. The key informant interviews were conducted using English language and the discussions took place in the offices of the providers. All the interviews were recorded manually and with a digital recorder. Personal contacts were made with all the participants after which a date for the interview was fixed.

All the twelve selected participants took part in the study. The interviews were conducted after working hours with the assistance of a note taker who summarized the responses of the providers in detailed notes. Follow up questions using probes were asked during the interviews so as to have a deeper understanding of any subject if the explanation was unclear. The average duration of the interviews was 41minutes.

## Data management

The recorded discussions of key informant interviews were transcribed verbatim following each session. For quality assurance purposes, the scripts were compared with the written notes for completeness and accuracy. Then each script was checked against the audiotape by an independent reviewer. As a way of verifying the quality of translations, tapes were doubly transcribed after which both scripts were checked for similarity and where differences existed, these were reconciled by the transcribers. Coding of transcripts was done based on themes as they emerged during the coding process. The themes from each interview were reviewed by the researcher and grouped under wider themes. QDA Miner Lite v2.0.6 was used in the analysis of the data.

Five themes emerged from the KII. They included utilization of antenatal and delivery services in health facilities including reasons for high and low utilizations, reasons women deliver at home and by traditional birth attendants and constraints to use of health facilities for antenatal and delivery services. Others included how to improve utilization of antenatal and delivery services and involvement of husbands in antenatal and delivery services.

## Ethical consideration

Ethical approval for the study was obtained from the Research and Ethics Committee of Ebonyi State University Abakaliki, Nigeria. Reference number, EBSU/DRIC/UREC/Vol. 04/064.

Participants were required to sign to a written informed consent form before the interview and the nature of the study, its importance and the level of their participation were made known to them. Participation in the study was voluntary and participants were assured that information provided during the interview will be kept confidential.

## Results

### Interviewer characteristics

The first author conducted all the interviews. She is a trained Nurse, (BSc Nursing) working at Alex Ekwueme Federal University Teaching Hospital Abakaliki, Nigeria. The interviews were part of the project for the award of Master of Public Health degree of Ebonyi State University Abakaliki, Nigeria. She was guided and supervised all through the project including the conduct of the key informant interviews by a Senior Lecturer in the Department of Community Medicine, College of Health Sciences of Ebonyi State University Abakaliki, Nigeria.

### Participants' profile

The age range of the participants was 35 to 53 years with a median age of 47 years. The years of experience of the discussants ranged from 8 to 24 years. Nine of the discussants were officers-in-charge of primary health centers, two were chief nursing officers of a tertiary health institution and a mission hospital while one is a Medical Officer in-charge of a General hospital. Six of the discussants have been in their current positions for 3 years and more. Eleven of the discussants were females and half of the discussants serve in the urban area.

### Utilization of antenatal and delivery services in health facilities

Most of the participants in urban area were of the opinion that there was good utilization of health facilities for antenatal and delivery services. All the participants pointed out that health facilities are utilized more for antenatal care than for delivery services. One of the participants made a graphic description by remarking that only about 30–40% of those who attend antenatal care in health facilities deliver in the same facilities. She ranked utilization of antenatal care as good while she was of the opinion that use of health facilities for delivery services was fair. In the rural area, half of the participants regarded the use of antenatal care and delivery services as good. All the participants also agreed that the use of antenatal care by the women is higher than that for delivery services.

### Reasons for good utilization of health facilities for antenatal and delivery services

Most providers in urban and rural areas that attested to good utilization of antenatal and delivery services in their respective facilities attributed this to the quality of health services being provided in the facilities. This concept of quality of care was expressed by the health workers in several ways. A participant in the urban area had this to say:

> *"We offer quality healthcare to our clients, we ensure that they are satisfied with the services they receive from us. We have also ensured that the attitude of health workers to the women is good"*

(Female participant, urban)

Another participant from the urban viewed it from the perspective of respectful maternity care. She presented her views this way:

*"We practice respectful maternity care. The clients are allowed to stay at any position that is comfortable to them during labour unlike before when health workers shunned such practices. Health workers now respect the dignity of the women and this creates a very good impression on them"*

(Female participant, urban)

A participant from the rural area perceived the rendering of good services to the clients as the main reason for the good utilization of antenatal and delivery services. She expressed it this way:

*"Good market sells itself. We attend to the women very well when they come and when they go home, they inform others of our good services. We have realized that the women cannot attend antenatal care or deliver in health facilities where they are harassed or treated less as human beings"*

(Female participant, rural)

One participant from the rural area attributed the good utilization in her facility to the fact that the facility offers a 24-hour service delivery. A participant from the urban area was of the opinion that a certain program which was on-going in her facility was the reason her facility was well utilized for antenatal and delivery services. She made known her thoughts this way:

*"There is a program we have in our hospital that is called "Hello Mama" **T**he aim is to reduce maternal death. It is all about messages and calls to the pregnant women including their husbands. The program came with incentives most times which attracted other women. And the messages inform them of what they need to know about pregnancy hence very good"*

(Female participant, urban)

### Reasons for poor utilization of health services for antenatal and delivery services

The participants in urban and rural areas gave different reasons for poor utilization of formal health facilities for antenatal and delivery services. Most of the providers in the urban area attributed the poor utilization of health facilities for antenatal and delivery services to poor health seeking behavior of the women while in the rural, it was attributed to the poor attitude of health workers. One of the participants in urban expressed her thoughts this way:

*"There is poor health seeking behavior among the people in this locality. They prefer to go to traditional birth attendants and patent medicine dealers for antenatal and delivery care and see the hospitals as a place to go only when there are complications"*

(Female participant, urban)

A participant in rural who was not speaking specifically for her health facility attributed the poor utilization to poor attitude of health workers. He had this to say:

*"The health workers are rarely seen on duty and when they come, their attitude discourages the women from coming again. The poor attitude of health workers encourages the women go to traditional birth attendants for services"*

(Male participant, rural)

Two participants each in urban and rural areas were however full of lamentations for the poor utilization of their respective health facilities for antenatal and delivery services. Their major grouse was the neglect of primary health centers by the women for maternal health services. These were how the participants expressed their views:

> *"This place (primary health center) does not meet up to an urban standard. People living in urban areas have choices as there are many big hospitals. People do not patronize primary health centers in urban areas because they are usually of low standard"*

(Female participant, urban)

> *"People here seem to value and prefer traditional birth attendants and patent medicine dealers to health centers for antenatal and delivery services. There are also those who would rather travel to Abakaliki, (the state capital) for such services thus abandoning the primary healthcare system"*

(Female participant, rural)

Another participant also from the rural area pointed out two sets of women who do not patronize primary health centers for antenatal and delivery services in the rural areas. The first are those who prefer a medical doctor attending to them during antenatal care while the other are those who perhaps by reason of cost prefer to go to traditional birth attendants or even deliver at home.

## Reasons women deliver at home and by traditional birth attendants

Most of the participants in urban and rural areas were of the opinion that ignorance is the major reason women deliver at home or with traditional birth attendants. This ignorance manifests in different forms. The participants had these to say:

> *"The women feel there is no need to come to hospital for delivery except when there are complications"*

(Female participant, urban)

> *"When a hospital is 'big' they conclude that the only thing they do is operation (Caesarean section) and based on that they are afraid of patronizing such health facilities"*

(Female participant, urban)

> *"They (the women) think that we charge too much but in truth we are not charging the way they think, it is just that they have made up their mind that our charges are on the high side"*

(Female participant, rural)

A few of the participants inferred that the women are influenced by community members especially women in their respective families and those who have delivered before on where to deliver. Unfortunately, they were of the opinion that most of the women they consult prefer the services of traditional birth attendants either because of their personal experiences or prejudice. These were how the participants expressed their thoughts:

*"The women are told by their mothers or relatives, that traditional birth attendants are the main place for delivery and that it is easier there (with TBA) than in hospitals where the health workers give unnecessary instructions"*

(Female participant, rural)

*"The women have confidence in the services of traditional birth attendants. They don't waste their time and they do not operate and many people especially mothers-in-law campaign for them in the communities"*

(Female participant, urban)

Two of the participants were of the opinion that culture also have a role to play as the women perceive delivery at home to be more natural than doing same in a hospital.

One of the participants in the urban however had a different opinion of delivery with traditional birth attendants. She gave the impression that delivery with traditional birth attendants is now a thing of the past. She made known her thoughts this way:

*"Any woman that comes for antenatal care in this facility cannot deliver at home or with traditional birth attendants. Even if they do not deliver in this health facility, they will do that either in a private hospital or another health center close to them because of the health education they receive during antenatal care"*

(Female participant, urban)

This view of every woman delivering in health facilities was supported by a participant in the rural area. She shared her experience though different this way:

*"There is one renowned traditional birth attendant in this community and she is now part of ward health committee so she does not provide services anymore. Since I have been transferred to this health center nobody has delivered with her. In-fact, she personally brought her neighbor who was in labour to this health center for delivery"*

(Female participant, rural)

## Constraints to use of health facilities for antenatal and delivery services

Most participants in urban and rural areas viewed ignorance as the main barrier to the use of health facilities for antenatal and delivery services. One of the participants in the rural area summarized it this way:

*"The women are not aware of the need to attend antenatal care and also deliver in health facilities. With the way things are at present, even if you make antenatal care free many women will still continue to deliver at home"*

(Female participant, rural)

Few participants ranked lack of finance as the next major barrier to the use of health facilities for antenatal and delivery services in urban and rural areas. A participant in rural area had this to say:

*"Lack of finance is a serious problem because despite the fact that we charge only a very little amount, some of the women cannot afford it. This is where the traditional birth attendants appeal to them as they allow them to pay their fees over a period of time"*

(Female participant, rural)

One participant in rural area inferred that the women have a wrong impression of the attitude of health workers which make them to patronize traditional birth attendants or deliver at home.

Some deficiencies of the health system which could have been regarded as barriers to utilization of services were viewed differently by the health workers as they have done their best to mitigate such circumstances. These were how the participants made known their views:

*"I am the only health worker in this health center and I do all the work. Even though I am overworked it does not affect round the clock service delivery in the health center as I reside in the health facility"*

(Female participant, rural)

*"The health center is not equipped, in-fact everything (meaning equipment), I am using here, I bought them with my own money. Equipment does not affect the delivery of services to our clients as we do the best with what we have"*

(Female participant, rural)

## How to improve utilization of antenatal and delivery services

Even though the participants did not specifically recognize the attitude of health workers as the major barrier to utilization of antenatal and delivery care in health facilities, they perceived that the health workers creating a good rapport with the women will help to improve utilization. This story was the same among participants in urban and rural areas. The views of the participants are summed up with these quotes:

*"The few women that come to health facilities should be treated with dignity and respect so that they go back happy and speak 'good' about the health facility. We (health workers) are the image makers of the health system and not just the facility where we are working. If we fail to 'retain' these women, they eventually become managed by quacks which is not good."*

(Female participant, urban)

*"There should be attitudinal change on the part of healthcare providers. Some providers may be tired and talk anyhow to the women and they go back with that impression and spread the information that health workers are rude. This will help form the opinion of other women about health workers."*

(Female participant, urban)

A few participants from the rural area were of the opinion that public enlightenment and home visits by health workers will go a long way in reinforcing the importance of utilizing health facilities for antenatal and delivery services to the women. One of the participants expressed her view this way:

*"Public enlightenment and home visits by health workers will help in emphasizing the relevance of antenatal care and delivery in health facilities to the women. If members of the community are included in the enlightenment, the impact will be great."*

(Female participant, rural)

One of the participants also from the rural area focused on the empowerment of women as very important especially when the husband does not appreciate the use of formal health facilities for antenatal care. She had this to say:

*"If a woman is educated and engaged in an economic activity, she may not wait for her husband or any other person to give her money before she attends antenatal care."*

(Female participant, rural)

## Involvement of husbands in antenatal and delivery services

All the participants in the two study groups agreed on the need to involve men in matters related to maternal healthcare. A participant in urban area pointed out the key role men play in family matters and the importance of such when applied to antenatal and delivery care. She made her views known this way:

*"A husband is the head of the home. Hardly will any man instruct his wife to go for antenatal care having provided money and other supports for that purpose and the woman refuses"*

(Female participant urban).

A few participants in the urban area traced the practical way they have involved men in matters related to antenatal care. One of the participants had this to say:

*"Just like in family planning, we also have male involvement in antenatal care. In our health facility, we attend to any woman that comes with her husband first during antenatal care. By this we make the husbands feel the importance of supporting their wives from the first day of pregnancy to delivery and this encourages the husbands to be part of antenatal care"*

(Female participant, urban)

A participant in the rural area collaborated this view however it appears the health workers there have not started putting this into practice. The participant made her views known this way:

*"We encourage men to assist their wives during pregnancy and to accompany them when they are attending antenatal clinics. This is because we have found out that it gives the women joy when their husbands accompany them to the health center for antenatal care"*

(Female participant, rural)

Another participant in the rural area pointed out the extra benefits that may accrue to the man in being part of antenatal care visits. This was how she made it known:

*"The antenatal care visits will give men the opportunity to learn from the health talks on how to care for their pregnant wives, save money towards the delivery of the baby and how the men could handle issues related to their own health"*

(Female participant, rural)

Another participant foresees another good in involving men in antenatal care issues. She presented her views this way.

*"The men should be involved in antenatal care so that some tests will be conducted for them. We have a programme in our facility where we send text messages and make calls to husbands encouraging them on how to take care of their wives during pregnancy"*

(Female participant, urban)

A participant in the urban area however explained that men are more involved during delivery than antenatal care. She gave a graphic description of that this way:

*"For delivery, let me say about 60–70% of men accompany their wives but when it comes to antenatal care less than 10% do that. In this facility, we give gifts and incentives to couples that attend antenatal care together"*

(Female participant, urban)

## Discussion

All the participants were of the opinion that use of health facilities for antenatal care is higher than that for delivery services in the study area. This is similar to findings from the Nigerian Demographic and Health Survey [4], in which the proportion of women utilizing antenatal care is higher than those that deliver in health facilities. It has also been observed that the use of health facilities for delivery services is poor [14, 18]. Most of the health workers in urban and rural areas attributed the good utilization of health facilities for maternal health services to the delivery of quality healthcare in the various health facilities. This concept of quality was described by the participants in various forms. This is commendable as women have been known to avoid using health facilities for maternal health services when they are not treated with respect.

An analysis of 2012 National HIV/AIDS and Reproductive Health Survey which involved individuals that did not utilize antenatal care revealed that improvement of quality of antenatal care was rated as the most essential factor towards improving its utilization after taking care of finance and other barriers [19]. To further emphasize the importance of quality maternal health service, a study in Zambia found that the main reason for low utilization of maternal health service was perceived low quality of services [20]. It is worth commenting that one of the providers in rural area viewed the delivery of round the clock services in her facility as being equivalent to good utilization of services. In a study in Enugu state, Nigeria, it was found that providers of maternal health services perceived good utilization of services as being same as quality healthcare delivery [21]. Observations such as these necessitated calls for health workers to be trained on quality of healthcare [14, 21].

The reasons for poor utilization of health facilities for antenatal and delivery services were different among participants in urban and rural areas. In urban, participants attributed that to

the poor health seeking behaviour of the women. In rural however attention was shifted to the poor attitude of health workers and the neglect of primary health centers which are the main health facilities in the area. It has been observed that efforts made to increase delivery at primary health centers is capable of decreasing the maternal mortality in rural areas of Nigeria [22]. Based on this, it has been postulated that in decreasing maternal mortality in Nigeria, attention should be given to rural areas and good utilization of primary health centers [14]. To attest to the neglect of primary health centers, a study in Enugu state, Nigeria, revealed that health centers were poorly utilized and that quality of care was also poor [23]. Such community perceptions of poor-quality care coupled with unavailable services further limit the utilization of primary health centers [24]. These feelings of poor service delivery from primary health centers could explain why most women move from rural to urban areas for antenatal and delivery services. Similarly, a study among Mesoamerican women revealed that there was a relationship between well-equipped health facilities and use of such facilities for delivery services. Thus, it was observed that women make long journeys to those facilities that were considered equipped [25]. This makes it imperative that good attention should be given to primary health centers and rural areas if the maternal death burden in Nigeria should be improved.

Most participants in urban and rural areas were of the opinion that ignorance is the main reason why women still deliver with traditional birth attendants and at home. It could be assumed that the providers were familiar with only one side of the divide. For example, in a study in northwest Nigeria, women identified ignorance, abuse and illiteracy as barriers to seeking delivery services in health facilities [26]. However, the same women attested that they patronize traditional birth attendants because their services were inexpensive and accessible [26]. This maybe an indication that there are factors that push the women away from delivering in health facilities and others that pull them to deliver with traditional birth attendants. This could account for the different reasons that make the women to patronize traditional birth attendants. For instance, in a study in southwest Nigeria, the major reasons for utilizing their services were that they were user friendly and accessible [27]. In a similar study in northwest Nigeria, women prefer delivery services from traditional birth attendants because of their positive attitude towards them and their services [28]. An influence of culture was observed in another study in northwest Nigeria, where the women preferred home delivery because of male attendants in health facilities and cost of services [29].

There are however good observations from the results of this study which point to a gradual and perhaps a possible end to the utilization of delivery services from traditional birth attendants. A participant in the urban area was certain no woman that attended antenatal care in her facility will deliver with a traditional birth attendant because of the health education the women receive during antenatal care. This remark is of good account as it has been found that women who make four or more antenatal care visits have increased odds of delivering in a health facility [7, 8]. Ensuring health facility delivery by the women is capable of improving maternal health in Nigeria. Among participants in the rural, one remarked that the involvement of a known traditional birth attendant in the ward health committee has been a boost to health facility delivery. This attests to the good report in community ownership and management of primary health centers and should be encouraged. These two approaches should be further explored towards increasing health facility delivery by the women especially in rural areas.

Elsewhere, several other reasons contribute towards encouraging women to deliver with traditional birth attendants and at home. Some of the findings are at variance with what was obtained from this study. For example, from a study in Ethiopia the main reason for utilizing the services of traditional birth attendants was because they considered health facility delivery as unnecessary and not part of their custom [30]. From the results of a study in Bangladesh,

poverty was the main reason for home delivery and with traditional birth attendants [31]. However, in western Kenya, delivery with traditional birth attendants was favoured because of the flexibility of payment for their services [32]. In another study in rural Kenya, home delivery was more as a result of health facilities being too far [33]. A combination of factors including physical distance from health facility and poverty prevented women from delivering with skilled providers in Indonesia [34].

Also, most providers in urban and rural areas perceived ignorance among the women as the major barrier to the use of antenatal and delivery services in health facilities. This is similar to findings from other geo-political zones in Nigeria though from different perspectives. For instance, from a study in south-south Nigeria it was found that women were unaware of maternal health services provided in health facilities [35]. This may make them not to appreciate the capacity and capability of health workers in formal health institutions for delivery services. In northern Nigeria, the need for increasing the awareness of the people on the relevance of utilizing health facilities for maternal health services was made [36]. In another study in south-south Nigeria, the need for comprehensive health education was emphasized and the aim was to let the women know the value of utilizing skilled birth attendants for delivery purposes [37]. A study in north-central Nigeria went beyond the women and concluded that the community as a whole lacked knowledge on the benefits of delivering in a health facility and as such the need to increase community awareness on the need for such services [29]. Thus, it is of importance to enlighten the public on the relevance of health facility delivery bearing in mind the high maternal death burden in Nigeria.

It could be said that this concept of ignorance concerning use of antenatal and delivery services is not restricted to Nigeria. In an analysis of the Demographic and Health Survey of countries in sub-Saharan Africa and south Asia, it was found that after taking care of service-related barriers, there was need to enlighten the public on the gains of utilizing health facilities for maternal health services [38]. In Pakistan, the results of a study revealed that there was lack of knowledge concerning antenatal care and this was perceived as the major barrier to utilizing antenatal care in health facilities [39]. In rural Kenya, it was found that women had poor understanding of antenatal care as it was seen as just a place for the treatment of common diseases [40]. Another study in Kenya revealed that there were misconceptions about care provided by health workers hence the need for a form of re-orientation of the women through health education [41]. Similarly, in Rajasthan it was found that changing the perception of women on antenatal care and delivery in health facilities is essential in increasing demand for such services by the women [42]. This was because the women were unaware of the importance of such services.

A few of the participants in urban and rural areas ranked lack of finance as the second barrier to utilizing health facilities for antenatal and delivery services. This observation is similar to what was obtained in Kenya, where cost was the major barrier to delivering in a health facility [32]. There is however a different finding from a study in Ghana where the major barrier to use of maternal health services was inadequate medical equipment and essential medicines [43]. Suffice it to say that participants in this study did not reckon with deficiencies associated with the health system as barriers to the utilization of maternal health services in health facilities. This shows that the providers of maternal healthcare go the extra mile to cover the weaknesses of the health system and this is commendable. It has also been found that certain societal ills could affect the utilization of maternal health services. In a study in southeast Nigeria, security challenges limited the use of maternal health services in primary health centers [21]. This was because most of the health centers were not fenced and based on security concerns the health workers refrained from opening the health centers for round the clock service delivery. To an extent, the health facility being open at all times is of good record and this may

have influenced the opinion of one of the participants in the rural area that her health center was well utilized for maternal health services since it was open at all times.

Most participants in urban and rural areas posited that to improve utilization, there should be a change in attitude of health workers to the women. This is contrary to their initial assertion that ignorance on the part of the women was the major barrier to the utilization of antenatal and delivery services in the study area. Perhaps the participants shied away from laying the blame on themselves. The issue of negative provider attitude among health workers is not new. From the results of a study in northern Nigeria, the main reason for non-utilization of health facilities for delivery services was negative provider attitude [36]. Elsewhere in Bayelsa state, Nigeria, it was found that health workers should encourage the utilization of maternal health services by having a positive attitude to women during visits to health facilities [35]. Aware of the need for a good provider attitude, another study in north central Nigeria concluded that antenatal care providers should be trained to improve the quality of care in the discharge of their duties [44]. This was perceived as a way of increasing utilization of health facilities for delivery services. Similarly, even though use of health facility deliveries were increasing in rural areas of Ghana, a major constraint in the use of such services was the maltreatment of the women by the health workers [45]. This was similar also to what was obtained in Kenya [32, 40].

Negative provider attitude by providers of maternal health care is at a great cost to the society. It has been ascertained that dissatisfaction with antenatal care by pregnant women could lead to preference for traditional based maternity care. Unfortunately, such actions are associated with maternal and fetal mortalities [46]. This necessitates that providers of maternal health services should provide quality maternal health service to their clients by having a good provider attitude. Realizing this feat may necessitate that health workers are trained on quality of care.

A few participants from rural area perceived that public enlightenment was very essential if the utilization of health facilities for antenatal and delivery care is to be increased. This may support the opinion of providers that ignorance is the most limiting factor to the use of health facilities for antenatal and delivery services. This finding is similar to what was obtained in Sokoto, Nigeria where the use of health promotion interventions was advocated so as to improve health facility delivery by women in rural area [47]. There has also been a call for community education aimed at increasing skilled attendant at delivery in rural Nigeria [48]. Even in studies outside Nigeria, there has been overwhelming calls to increase community awareness on the importance of utilizing health facilities for antenatal and delivery services [32, 49]. From a study in Nepal, it was observed that women had poor knowledge of services offered by skilled birth attendants [11]. In Indonesia, there was emphasis in maximizing health promotion efforts in increasing community awareness of using health services generally [34]. This was because use of health facilities for sundry services in that region was poor. Bearing in mind the high maternal death burden in Nigeria, there is need to increase community awareness on the utilization of health facilities for maternal health services. This will cater for the ignorance of the women on the use of such services. This should be complemented by training health workers on quality of care so that the continual use of health facilities for such services by the women could be guaranteed.

All the participants in urban and rural areas were in favour of the involvement of men in matters related to maternal health services. This was based on the useful role men could play in the affairs of the family. Thus, involvement of men has been viewed as a strong influence in the health of their partners and children [50]. Perhaps, because delivery is a one-time facility visit, it was observed that more men accompany their wives for delivery services than antenatal care. Suffice it to say that the need to involve men in issues related to maternal healthcare is of

relevance. For example, a study in rural Nigeria concluded that community elders should be supported to initiate rules that will promote the involvement of men in maternal health as this has the capacity of increasing the use of skilled attendants by women during delivery [51]. Similarly, another study in northern Nigeria advocated the involvement of men in matters related to delivery of maternal health services [52].

Similar observations have also been recorded in studies outside Nigeria. For instance, there has been a call to address the barriers in involving men in maternal health issues in Ghana [53]. Also, in Zambia, it was advocated that health education programmes should encourage male involvement in the decision of pregnant women to seek antenatal care so as to encourage adequate use of services [54]. These observations may indicate the need to involve men in use of health facilities for maternal health services as this may have a good role to play in reducing the maternal death burden in Nigeria.

## Conclusions

Participants were aware of values of good provider attitude and this is commendable. This combined with the finding of poor attitude of health workers necessitates that health workers are trained on quality of care. There is need for public enlightenment on importance of utilizing health facilities for antenatal and delivery services. Community ownership of primary health centers especially in rural communities will enhance utilization of such facilities and should be strengthened. Efforts should be made to involve men in matters related to maternal healthcare as this may have a positive influence in improving maternal health in Nigeria.

## Supporting information

**S1 File. Key informant interview Guide.**
(DOCX)

**S2 File.**
(DOCX)

## Author Contributions

**Conceptualization:** Pearl Chizobam Eke, Edmund Ndudi Ossai, Lawrence Ulu Ogbonnaya.

**Data curation:** Pearl Chizobam Eke, Irene Ifeyinwa Eze.

**Formal analysis:** Edmund Ndudi Ossai.

**Methodology:** Pearl Chizobam Eke, Edmund Ndudi Ossai, Irene Ifeyinwa Eze, Lawrence Ulu Ogbonnaya.

**Writing – original draft:** Edmund Ndudi Ossai.

**Writing – review & editing:** Pearl Chizobam Eke, Edmund Ndudi Ossai, Irene Ifeyinwa Eze, Lawrence Ulu Ogbonnaya.

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
