## [Decision Letter · Decision Letter 0]

30 Nov 2020

PONE-D-20-30363

Exploring providers’ perceived barriers to utilization of antenatal and delivery services in urban and rural communities of Ebonyi state, Nigeria: a qualitative study

PLOS ONE

Dear Dr. EDMUND NDUDI OSSAI,

Thank you for submitting your manuscript to PLOS ONE. After careful consideration, we feel that it has merit but does not fully meet PLOS ONE’s publication criteria as it currently stands. Therefore, we invite you to submit a revised version of the manuscript that addresses the points raised during the review process.

We look forward to receiving your revised manuscript.

Kind regards,

Sharon Mary Brownie

Academic Editor

PLOS ONE

Additional Editor Comments 

The review process has identified very significant issues that require attention within your manuscript if it is to be considered for publication. Of concern, your qualitative quotes do not support your findings. For a returned manuscript please ensure that you include a completed 32 step COREQ checklist of the quality reporting of qualitative research https://www.equator-network.org/reporting-guidelines/coreq/ Please also include a table addressing each point raised by the reviewers.

Journal Requirements:

3. When reporting the results of qualitative research, we suggest consulting the COREQ guidelines: http://intqhc.oxfordjournals.org/content/19/6/349. In this case, please consider including more information on the number of interviewers, their training and characteristics.Moreover, please provide the interview guide as a Supplementary File.

Reviewers' comments:

Reviewer's Responses to Questions

**Comments to the Author**

1. Is the manuscript technically sound, and do the data support the conclusions?

Reviewer #1: Partly

Reviewer #2: Yes

2. Has the statistical analysis been performed appropriately and rigorously? 

Reviewer #1: N/A

Reviewer #2: N/A

3. Have the authors made all data underlying the findings in their manuscript fully available?

Reviewer #1: Yes

Reviewer #2: Yes

4. Is the manuscript presented in an intelligible fashion and written in standard English?

Reviewer #1: Yes

Reviewer #2: Yes

5. Review Comments to the Author

Reviewer #1: Introduction

• Much of the literature in the introduction focus on maternal deaths, and access and utilization of ANC as mitigations. Authors worked their way deductively from global perspectives to the Nigerian situation. However, the purpose of the study is to explore providers’ perceived barriers to utilization of antenatal and delivery services in urban and rural Nigerian state. Accordingly, authors should situate the study within theories underlying barriers to utilization of ANC and delivery services.

• What exactly is the problem underlying this study? what does this study seek to achieve? How does this study differ from the vast literatures addressing gaps in maternal and child health within and across LMICs including Nigeria?.

• The conceptualization in its current form is not grounded in new and innovative theoretical perspectives beyond what is known on this subject matter.

Methodology

• Authors stated: “In the second stage, a list of all health facilities in the selected local government areas was made and ranked based on the number of women that utilized the health facilities for antenatal and delivery services in the last six months based on hospital records. Twelve health facilities, six each in urban and rural communities were selected based on this criterion”. It is unclear what the criterion is and how it aided the selection.

• Authors stated: “The key providers of antenatal and delivery services in each of the facilities were purposively selected for interview”. Who were these key providers and why were they sampled purposively?

• Several content overlaps and repetitions appear though out the manuscript and especially the methodology. This phrase “pretested interview guide”, for example is overly repeated throughout the methodology.

Findings

• The lack of systematic conceptualization of perceived barriers from the onset has affected the quality of the findings – which are a bit disjointed. Example, thematic areas of the findings included: Utilization of antenatal and delivery services in health facilities; Reasons for good utilization of health facilities for antenatal and delivery services; Reasons for poor utilization of health services for antenatal and delivery services; Reasons women deliver at home and by traditional birth attendants; Constraints to use of health facilities for antenatal and delivery services; and How to improve utilization of antenatal and delivery services.

• I find it extremely difficult collectively relating the thematic areas to barriers of ANC and delivery service use.

• Moreover, authors earlier argued that quality of maternal of health services in the study areas is extremely weak. This tend to deter women from utilizing ANC and subsequent uptake of delivery services. To the contrary, the first set of findings suggest that providers are doing their utmost best to provide good quality maternal health services. This reflect in these quotes below:

“We offer quality healthcare to our clients, we ensure that they are satisfiedbwith the services they receive from us. We have also ensured that the attitude of health workers is good” (Female participant, urban)

We practice respectful maternity care. The clients are allowed to stay at any position that is comfortable to them during labour unlike before when health workers shunned such practices. Health workers now respect the dignity of the women and this creates a very good impression on them” (Female participant, urban)

• It can be inferred from the quotes above that authors are evading the purpose of this study. Further, to say in one breath that providers are oriented to good quality care and to turn around and accused the same providers of poor attitude appears to turn this study upside down.

Other comments

While authors have done their best in coming out with this manuscript, the conceptualization, findings and conclusions present no significant new information. There are a plethora of quality theories and evidence produced in response to barriers of maternal and delivery service utilization in sub-Saharan Africa. Such findings produced barriers that cut across poverty, income, demographic and geographic factors, cultural and social norms, religion and many more. You can read more about these studies at: https://scholar.google.com/scholar?start=20&q=why+women+deliver+at+home&hl=en&as_sdt=0,5 and particularly the work of Mrisho and colleagues (2007).

Reviewer #2: The authors did a good job presenting the barriers around severe maternal mortality in the southeastern part of Nigeria. While the study falls short of a large sample size, it reveals some of the barriers that can be acted upon as helpful and much-needed steps towards a positive intervention for reducing severe maternal mortality in that part of the country.

I have some minor suggested changes, though.

1. The authors could be silent on the majority Christian mentioned in the method section, except if their study shows that religiosity plays a role in the health outcomes.

2. In the study design section, correct the punctuation typo before (KII).

3. In the study instrument, rephrase the sentence to remove "also" at the end of "use of recorders also.".

4. Combine sections "Study Instrument, Data Collection Methods, Pretesting of instrument" into a single section. There seem to be some repetitions and commonalities.

6. PLOS authors have the option to publish the peer review history of their article (what does this mean?). If published, this will include your full peer review and any attached files.

Reviewer #1: No

Reviewer #2: No

---

## [Author Response · Author response to Decision Letter 0]

10 Feb 2021

PONE-D-20-30363

Exploring providers’ perceived barriers to utilization of antenatal and delivery services in urban and rural communities of Ebonyi state, Nigeria: a qualitative study

PLOS ONE

Dear Dr. EDMUND NDUDI OSSAI,

Thank you for submitting your manuscript to PLOS ONE. After careful consideration, we feel that it has merit but does not fully meet PLOS ONE’s publication criteria as it currently stands. Therefore, we invite you to submit a revised version of the manuscript that addresses the points raised during the review process.

We look forward to receiving your revised manuscript.

Kind regards,

Sharon Mary Brownie

Academic Editor

PLOS ONE

Additional Editor Comments 

The review process has identified very significant issues that require attention within your manuscript if it is to be considered for publication. Of concern, your qualitative quotes do not support your findings. For a returned manuscript please ensure that you include a completed 32 step COREQ checklist of the quality reporting of qualitative research https://www.equator-network.org/reporting-guidelines/coreq/ Please also include a table addressing each point raised by the reviewers.

Journal Requirements:

 Author response: This has been complied with.

 Author response: Thanks for the observation

The manuscript has been reviewed by a senior colleague and research partner, Dr. Chinawa Josephat N.

He is an Associate Professor of Paediatrics, Department of Paediatrics, College of Medicine, University of Nigeria, Enugu Campus

3. When reporting the results of qualitative research, we suggest consulting the COREQ guidelines: http://intqhc.oxfordjournals.org/content/19/6/349. In this case, please consider including more information on the number of interviewers, their training and characteristics. Moreover, please provide the interview guide as a Supplementary File.

 Author response: Thanks for the guidance. The manuscript you referred me to was remarkably good, very helpful and has become a good source of reference.

 Author response: The authors have requested for assistance in this regard bearing in mind that this was a qualitative study. Depositing such materials is easier with records from a quantitative data collection method. The authors will be pleased to know what we are to submit and how it will be done. We need your assistance, Sir.

Reviewers' comments:

Reviewer's Responses to Questions

Comments to the Author

1. Is the manuscript technically sound, and do the data support the conclusions?

Reviewer #1: Partly

Reviewer #2: Yes

2. Has the statistical analysis been performed appropriately and rigorously?

Reviewer #1: N/A

Reviewer #2: N/A

3. Have the authors made all data underlying the findings in their manuscript fully available?

Reviewer #1: Yes

Reviewer #2: Yes

4. Is the manuscript presented in an intelligible fashion and written in standard English?

Reviewer #1: Yes

Reviewer #2: Yes

5. Review Comments to the Author

Reviewer #1: Introduction

• Much of the literature in the introduction focus on maternal deaths, and access and utilization of ANC as mitigations. Authors worked their way deductively from global perspectives to the Nigerian situation. However, the purpose of the study is to explore providers’ perceived barriers to utilization of antenatal and delivery services in urban and rural Nigerian state. Accordingly, authors should situate the study within theories underlying barriers to utilization of ANC and delivery services.

• What exactly is the problem underlying this study? what does this study seek to achieve? How does this study differ from the vast literatures addressing gaps in maternal and child health within and across LMICs including Nigeria?.

• The conceptualization in its current form is not grounded in new and innovative theoretical perspectives beyond what is known on this subject matter.

Author response: Thanks for the observation. A paragraph has been included that focused on providers perspectives on barriers to delivery of antenatal and delivery services.

Methodology

• Authors stated: “In the second stage, a list of all health facilities in the selected local government areas was made and ranked based on the number of women that utilized the health facilities for antenatal and delivery services in the last six months based on hospital records. Twelve health facilities, six each in urban and rural communities were selected based on this criterion”. It is unclear what the criterion is and how it aided the selection.

Author response: The aim here was to select health facilities which were mostly patronized by the women for antenatal and delivery services. Based on the level of utilization, it is expected that this will impact on the experience of the providers. The authors were of the opinion that the views of the providers in these health facilities that are well utilized will be more useful than those in health facilities not well utilized. 

Authors stated: “The key providers of antenatal and delivery services in each of the facilities were purposively selected for interview”. Who were these key providers and why were they sampled purposively?

Author response: Twelve providers pf antenatal and delivery services participated in the study. Nine were Officers-in-Charge of the primary health centers. Two were Chief Nursing Officers of a tertiary health institution and a mission hospital. One participant was a Medical Officer in charge of a General hospital. 

The selection of the providers was based on their official positions and the roles they play in policy formulation and in the delivery of maternal health services in the health facilities. 

• Several content overlaps and repetitions appear though out the manuscript and especially the methodology. This phrase “pretested interview guide”, for example is overly repeated throughout the methodology.

Author response: Thanks for the observation. Corrections have been made. 

Findings

• The lack of systematic conceptualization of perceived barriers from the onset has affected the quality of the findings – which are a bit disjointed. Example, thematic areas of the findings included: Utilization of antenatal and delivery services in health facilities; Reasons for good utilization of health facilities for antenatal and delivery services; Reasons for poor utilization of health services for antenatal and delivery services; Reasons women deliver at home and by traditional birth attendants; Constraints to use of health facilities for antenatal and delivery services; and How to improve utilization of antenatal and delivery services.

Author response: These were themes that emerged during the coding process. The authors explored the views of the health providers on issues related to antenatal and delivery services in the study area.

• I find it extremely difficult collectively relating the thematic areas to barriers of ANC and delivery service use.

Author response: Barriers to antenatal and delivery services was one of the themes and perhaps the main subject of the interview/ study.

• Moreover, authors earlier argued that quality of maternal of health services in the study areas is extremely weak. This tend to deter women from utilizing ANC and subsequent uptake of delivery services. To the contrary, the first set of findings suggest that providers are doing their utmost best to provide good quality maternal health services. This reflect in these quotes below:

“We offer quality healthcare to our clients, we ensure that they are satisfied with the services they receive from us. We have also ensured that the attitude of health workers is good” (Female participant, urban)

We practice respectful maternity care. The clients are allowed to stay at any position that is comfortable to them during labour unlike before when health workers shunned such practices. Health workers now respect the dignity of the women and this creates a very good impression on them” (Female participant, urban)

• It can be inferred from the quotes above that authors are evading the purpose of this study. Further, to say in one breath that providers are oriented to good quality care and to turn around and accused the same providers of poor attitude appears to turn this study upside down.

Author response: Thanks for these comments. Not all the health providers asserted that their health facilities were well utilized for antenatal and delivery services. When it came to constraints, the health workers spoke generally perhaps based on their experiences and other aspects. One remarkable conclusion of this study is the need to train health workers in the study area on quality of care. Indeed, the health workers are doing their best but that best is not good enough.

 It has already been observed that the health workers have a poor perception of quality of care.

Ossai EN, Uzochukwu BSC. Providers’ perception of quality of care and constraints to delivery of quality maternal health services in primary health centers of Enugu state. Nigeria. IJTDH. 2015;8(1):13-24.

The health workers are aware that the women desire quality healthcare. Thus, it may be described as good that in all, the providers were also aware that they (the providers) are not doing enough in taking care of the women. This necessitates the need to train the health workers on quality of care.

Other comments

While authors have done their best in coming out with this manuscript, the conceptualization, findings and conclusions present no significant new information. There are a plethora of quality theories and evidence produced in response to barriers of maternal and delivery service utilization in sub-Saharan Africa. Such findings produced barriers that cut across poverty, income, demographic and geographic factors, cultural and social norms, religion and many more. You can read more about these studies at: https://scholar.google.com/scholar?start=20&q=why+women+deliver+at+home&hl=en&as_sdt=0,5 and particularly the work of Mrisho and colleagues (2007).

Author response: Thanks for this information. However, the authors in this study were concerned with the perspective of the health providers on the barriers to antenatal and delivery services in the study area. It may be expected that the health workers will concentrate more on health system factors. 

Reviewer #2: The authors did a good job presenting the barriers around severe maternal mortality in the southeastern part of Nigeria. While the study falls short of a large sample size, it reveals some of the barriers that can be acted upon as helpful and much-needed steps towards a positive intervention for reducing severe maternal mortality in that part of the country.

Author response: Thanks very much for this observation.

I have some minor suggested changes, though.

1. The authors could be silent on the majority Christian mentioned in the method section, except if their study shows that religiosity plays a role in the health outcomes.

Author response: This has been corrected.

2. In the study design section, correct the punctuation typo before (KII).

Author response: This has been corrected.

3. In the study instrument, rephrase the sentence to remove "also" at the end of "use of recorders also.".

Author response: Corrected.

4. Combine sections "Study Instrument, Data Collection Methods, Pretesting of instrument" into a single section. There seem to be some repetitions and commonalities.

Author response: Corrected.

6. PLOS authors have the option to publish the peer review history of their article (what does this mean?). If published, this will include your full peer review and any attached files.

Author response: Yes. The peer review history can be published.

Do you want your identity to be public for this peer review? For information about this choice, including consent withdrawal, please see our Privacy Policy.

Reviewer #1: No

Reviewer #2: No

---

## [Decision Letter · Decision Letter 1]

23 Mar 2021

PONE-D-20-30363R1

Exploring providers’ perceived barriers to utilization of antenatal and delivery services in urban and rural communities of Ebonyi state, Nigeria: a qualitative study

PLOS ONE

Dear Dr. EDMUND NDUDI OSSAI,

Thank you for submitting your manuscript to PLOS ONE. After careful consideration, we feel that it has merit but does not fully meet PLOS ONE’s publication criteria as it currently stands. Therefore, we invite you to submit a revised version of the manuscript that addresses the points raised during the review process.

We look forward to receiving your revised manuscript.

Kind regards,

Sharon Mary Brownie

Academic Editor

PLOS ONE

Journal Requirements:

Additional Editor Comments

The manuscript is much improved in the revised version, however, reviewers have identified the need for a number of further improvements. Please consider each carefully and respond to these in full.

Reviewers' comments:

Reviewer's Responses to Questions

**Comments to the Author**

Reviewer #2: All comments have been addressed

Reviewer #3: (No Response)

Reviewer #4: All comments have been addressed

Reviewer #5: All comments have been addressed

2. Is the manuscript technically sound, and do the data support the conclusions?

Reviewer #2: Yes

Reviewer #3: Yes

Reviewer #4: Yes

Reviewer #5: Yes

3. Has the statistical analysis been performed appropriately and rigorously? 

Reviewer #2: Yes

Reviewer #3: Yes

Reviewer #4: N/A

Reviewer #5: N/A

4. Have the authors made all data underlying the findings in their manuscript fully available?

Reviewer #2: Yes

Reviewer #3: Yes

Reviewer #4: Yes

Reviewer #5: Yes

5. Is the manuscript presented in an intelligible fashion and written in standard English?

Reviewer #2: Yes

Reviewer #3: Yes

Reviewer #4: Yes

Reviewer #5: Yes

6. Review Comments to the Author

Reviewer #2: The authors have addressed all comments. Exploring providers’ perceived barriers to utilization of antenatal and delivery services could help reduce severe maternal mortality in that part of the country.

Reviewer #3: 1. The research question clearly seeks opinion on "barriers" for the utilisation of the Health Facility, whereas the results begin highlighting that the services are well utilised. The paper then goes on to explain how the services rendered can be made more accessible and better but the link to the reader breaks at the first level.

2. What Framework was used for the analysis of the qualitative data?

3. Some recommendations to the barriers can be added in the conclusion for a stronger way forward.

Reviewer #4: The article is very well written and I enjoyed reading the discussion and comments of reviewers have been addressed

Reviewer #5: The authors have sought to explore the reasons for barriers for utilization of antenatal and delivery services from the care-providers' perspective in some rural and urban communities of Ebonyi, Nigeria, through this qualitative study. In reply to previous reviewer comment, they have now amply explained the reasons for choosing the particular centers and the providers for interview. However, they have not been able to adequately answer, what new knowledge has this study added to existing literature because the reasons for not attending for antenatal care and delivery at health-care centers in LMICs are already well-known through previous literature. A few more points on how this could be improved would be beneficial add to knowledge.

There is a lot of repetition in the manuscript (discussion) regarding reasons for poor utilisation of health services. This could be stated more concisely.

7. PLOS authors have the option to publish the peer review history of their article (what does this mean?). If published, this will include your full peer review and any attached files.

Reviewer #2: No

Reviewer #3: No

Reviewer #4: No

Reviewer #5: No

---

## [Author Response · Author response to Decision Letter 1]

24 Apr 2021

Response to Reviewers’

Editor’s comment

Journal Requirements:

Author response

Thank you very much for this observation. The authors were not very sure of the very reference cited in this manuscript that has been retracted. However, the authors reviewed the whole references included in this manuscript and removed reference numbers 3, 5 and 6. They were replaced with new references.

We are hopeful that we have resolved this very important issue by this action we have taken. We also plead that we should be guided by the Editor to have this problem resolved assuming our review of the references was not of good effect.

Additional Editor Comments

The manuscript is much improved in the revised version, however, reviewers have identified the need for a number of further improvements. Please consider each carefully and respond to these in full.

Reviewers' comments:

Reviewer's Responses to Questions

Comments to the Author

Reviewer #2: All comments have been addressed

Reviewer #3: (No Response)

Reviewer #4: All comments have been addressed

Reviewer #5: All comments have been addressed

2. Is the manuscript technically sound, and do the data support the conclusions?

Reviewer #2: Yes

Reviewer #3: Yes

Reviewer #4: Yes

Reviewer #5: Yes

3. Has the statistical analysis been performed appropriately and rigorously?

Reviewer #2: Yes

Reviewer #3: Yes

Reviewer #4: N/A

Reviewer #5: N/A

4. Have the authors made all data underlying the findings in their manuscript fully available?

Reviewer #2: Yes

Reviewer #3: Yes

Reviewer #4: Yes

Reviewer #5: Yes

5. Is the manuscript presented in an intelligible fashion and written in standard English?

Reviewer #2: Yes

Reviewer #3: Yes

Reviewer #4: Yes

Reviewer #5: Yes

6. Review Comments to the Author

Reviewer’s comment 

Reviewer #2: The authors have addressed all comments. Exploring providers’ perceived barriers to utilization of antenatal and delivery services could help reduce severe maternal mortality in that part of the country.

Author response

Thanks for your kind comment

Reviewer #3: 1. The research question clearly seeks opinion on "barriers" for the utilisation of the Health Facility, whereas the results begin highlighting that the services are well utilised. The paper then goes on to explain how the services rendered can be made more accessible and better but the link to the reader breaks at the first level.

Reviewer’s comment

2. What Framework was used for the analysis of the qualitative data?

Author response

This was a qualitative study and the authors have stated that coding was based on themes as they emerged during the coding process.

However, in terms of a framework, the concept was to first enquire about the rating of the selected health facilities in terms of utilization of antenatal and delivery services and reasons for such classifications based on the views of the providers. Then the reasons why women deliver at home and with traditional birth attendants. Then the concept of barriers to use of formal health facilities for antenatal and delivery services.

We did not stop at this point because bearing in mind that Nigeria bears the greatest burden of maternal deaths globally, the authors sought to find out what could be done to improve the utilization of antenatal and delivery services. This was to serve as the ways of reversing the trend especially by policy makers. We also sought to know the effects of involving husbands in matters related to antenatal and delivery services from the view point of health providers.

Reviewer’s comments

3. Some recommendations to the barriers can be added in the conclusion for a stronger way forward.

Author response 

The recommendations have been included.

Thanks.

Reviewer’s comment

Reviewer #4: The article is very well written and I enjoyed reading the discussion and comments of reviewers have been addressed

Author’s response

Thanks for your kind consideration

Reviewer’s comment

Reviewer #5: The authors have sought to explore the reasons for barriers for utilization of antenatal and delivery services from the care-providers' perspective in some rural and urban communities of Ebonyi, Nigeria, through this qualitative study. In reply to previous reviewer comment, they have now amply explained the reasons for choosing the particular centers and the providers for interview. However, they have not been able to adequately answer, what new knowledge has this study added to existing literature because the reasons for not attending for antenatal care and delivery at health-care centers in LMICs are already well-known through previous literature. A few more points on how this could be improved would be beneficial add to knowledge.

There is a lot of repetition in the manuscript (discussion) regarding reasons for poor utilisation of health services. This could be stated more concisely.

Author response

Thanks for the observations.

The authors explored the views of the providers of maternal health services on the barriers to use of antenatal and delivery services and how to improve the utilization of such services. This is a very important problem in the locality bearing in mind the poor maternal health index in Nigeria. This makes the problems identified as being specific to the locality hence very relevant. It could then be said that the recommendations from this study if applied could be of good effect.

---

## [Decision Letter · Decision Letter 2]

10 May 2021

Exploring providers’ perceived barriers to utilization of antenatal and delivery services in urban and rural communities of Ebonyi state, Nigeria: a qualitative study

PONE-D-20-30363R2

Dear Dr. EDMUND NDUDI OSSAI,

We’re pleased to inform you that your manuscript has been judged scientifically suitable for publication and will be formally accepted for publication once it meets all outstanding technical requirements.

Kind regards,

Sharon Mary Brownie

Academic Editor

PLOS ONE

Reviewers' comments:

Reviewer's Responses to Questions

**Comments to the Author**

1. If the authors have adequately addressed your comments raised in a previous round of review and you feel that this manuscript is now acceptable for publication, you may indicate that here to bypass the “Comments to the Author” section, enter your conflict of interest statement in the “Confidential to Editor” section, and submit your "Accept" recommendation.

Reviewer #2: All comments have been addressed

Reviewer #5: All comments have been addressed

2. Is the manuscript technically sound, and do the data support the conclusions?

Reviewer #2: Yes

Reviewer #5: Yes

3. Has the statistical analysis been performed appropriately and rigorously? 

Reviewer #2: Yes

Reviewer #5: N/A

4. Have the authors made all data underlying the findings in their manuscript fully available?

Reviewer #2: Yes

Reviewer #5: Yes

5. Is the manuscript presented in an intelligible fashion and written in standard English?

Reviewer #2: Yes

Reviewer #5: Yes

6. Review Comments to the Author

Reviewer #2: The authors have addressed all reviewers' questions. The study is quite important to a region of Africa with high prevalence of maternal mortality.

Reviewer #5: The authors have answered all reviewer queries very well and the article is now good enough to be accepted for publication.

7. PLOS authors have the option to publish the peer review history of their article (what does this mean?). If published, this will include your full peer review and any attached files.

Reviewer #2: No

Reviewer #5: **Yes: **Azra Amerjee

---

## [Editor Report · Acceptance letter]

12 May 2021

PONE-D-20-30363R2 

Exploring providers’ perceived barriers to utilization of antenatal and delivery services in urban and rural communities of Ebonyi state, Nigeria: a qualitative study 

Dear Dr. Ossai:

I'm pleased to inform you that your manuscript has been deemed suitable for publication in PLOS ONE. Congratulations! Your manuscript is now with our production department. 

Kind regards, 

on behalf of

Professor Sharon Mary Brownie 

Academic Editor

PLOS ONE